# Ketogenic diet reduces early mortality following traumatic brain injury in *Drosophila* via the PPARγ ortholog Eip75B

Joseph Blommer[1], Megan C. Fischer[1], Athena R. Olszewski[1], Rebeccah J. Katzenberger[1], Barry Ganetzky[2], David A. Wassarman[1] *

**1** Department of Medical Genetics, School of Medicine and Public Health, University of Wisconsin-Madison, Madison, Wisconsin, United States of America, **2** Department of Genetics, College of Agricultural and Life Sciences, University of Wisconsin-Madison, Madison, Wisconsin, United States of America

* dawassarman@wisc.edu

**Data Availability Statement:** All relevant data are within the manuscript.

## Abstract

Traumatic brain injury (TBI) is a common neurological disorder whose outcomes vary widely depending on a variety of environmental factors, including diet. Using a *Drosophila melanogaster* TBI model that reproduces key aspects of TBI in humans, we previously found that the diet consumed immediately following a primary brain injury has a substantial effect on the incidence of mortality within 24 h (early mortality). Flies that receive equivalent primary injuries have a higher incidence of early mortality when fed high-carbohydrate diets versus water. Here, we report that flies fed high-fat ketogenic diet (KD) following TBI exhibited early mortality that was equivalent to that of flies fed water and that flies protected from early mortality by KD continued to show survival benefits weeks later. KD also has beneficial effects in mammalian TBI models, indicating that the mechanism of action of KD is evolutionarily conserved. To probe the mechanism, we examined the effect of KD in flies mutant for Eip75B, an ortholog of the transcription factor PPARγ (peroxisome proliferator-activated receptor gamma) that contributes to the mechanism of action of KD and has neuroprotective effects in mammalian TBI models. We found that the incidence of early mortality of *Eip75B* mutant flies was higher when they were fed KD than when they were fed water following TBI. These data indicate that Eip75B/PPARγ is necessary for the beneficial effects of KD following TBI. In summary, this work provides the first evidence that KD activates PPARγ to reduce deleterious outcomes of TBI and it demonstrates the utility of the fly TBI model for dissecting molecular pathways that contribute to heterogeneity in TBI outcomes.

## Introduction

Traumatic brain injury (TBI) is a major health issue worldwide [1]. It is a leading cause of disability and death, and its clinical management is challenging because the physical, behavioral, cognitive, and emotional sequelae are highly variable. Variation in sequelae among TBI patients stems from heterogeneity of primary injuries to the brain as well as heterogeneity of

**Funding:** Research reported in this publication was supported by the National Institute of Neurological Disorders and Stroke of the National Institutes of Health (grant number RF1NS114359) awarded to DAW. The content is solely the responsibility of the authors and does not necessarily represent the official views of the National Institutes of Health. JB was supported by a Sophomore Research Fellowship and a Hilldale Research Fellowship from UW-Madison. MCF was supported by a UW-Madison Genetics Department Summer Fellowship. The funders had no role in study design, data collection and analysis, decision to publish, or preparation of the manuscript.

**Competing interests:** The authors have declared that no competing interests exist.

genetic and environmental factors such as physical activity, sleep, and diet that promote tissue repair or exacerbate tissue damage through secondary injury mechanisms [2, 3]. Cellular and molecular mechanisms associated with secondary injuries include ionic imbalance, excitotoxicity, oxidative stress, inflammation, and mitochondrial dysfunction that disrupt cellular metabolism leading to neuronal dysfunction and cell death [4].

Glucose is the main energy source for the brain, but following TBI, glucose uptake and use by the brain is progressively reduced [5, 6]. Under these circumstances, ketone bodies, derived from fatty acid oxidation in the liver, become the main energy source for the brain [7, 8]. Ketone bodies such as β-hydroxybutyrate, acetone, and acetoacetate improve mitochondrial metabolism, reduce production of reactive oxygen species and proinflammatory proteins, and have broad neuroprotective effects [8, 9]. Elevated levels of ketone bodies in the blood, a state known as ketosis, can be induced by fasting and by high-fat, low-carbohydrate, low-protein ketogenic diet (KD). KD reduces seizures in refractory childhood epilepsy and ameliorates detrimental outcomes in mammalian models of neurological disorders, including TBI [10, 11]. In rat TBI models, KD reduces apoptosis, contusion volumes, and anxiety- and depressive-like behaviors and improves motor and cognitive performance [12–18]. However, much remains to be learned about the influence of KD in TBI, including the extent to which genetic background modulates its beneficial effects.

The ligand-dependent transcription factor PPARγ (peroxisome proliferator-activated receptor gamma) contributes to the mechanism of action of KD and has anti-inflammatory and neuroprotective effects in mammalian models of neurological disorders, including TBI [19, 20]. Activation of PPARγ by fatty acids inhibits inflammation by a variety of mechanisms, including by reducing the activity of nuclear factor-kappa B (NF-κB) transcription factors that promote expression of inflammatory genes encoding cytokines, chemokines, and adhesion molecules [21]. In rodent TBI models, the PPARγ agonist pioglitazone is protective against mitochondrial dysfunction, cognitive impairment, cortical tissue loss, inflammation, dendritic morphological changes, and long-term memory loss [22–25]. However, it is not yet known if PPARγ mediates the beneficial effects of KD in TBI.

To investigate the role of genetic and environmental factors in TBI outcomes, we developed a *Drosophila melanogaster* model of TBI [26, 27]. The fly TBI model uses a High-Impact Trauma (HIT) device to deliver blunt force injuries to the head and body of unanesthetized flies. Behavioral outcomes of TBI shared between flies and humans include temporary incapacitation, ataxia, abnormal sleep, early mortality, and reduced lifespan [26–31]. Cellular and molecular outcomes are also shared, including progressive neurodegeneration, disruption of the blood-brain barrier and the intestinal barrier, transient hyperglycemia, and prolonged activation of innate immune response pathways [26, 28, 29, 32, 33]. Using the Mortality Index at 24 hours ($MI_{24}$)—the normalized percent of flies that die within 24 h after strikes from the HIT device—as a readout, we previously found that genetic background plays a substantial role in determining TBI outcomes. For example, the $MI_{24}$ of flies injured at 0–7 days old varies from 7 to 58 among 179 inbred lines in the Drosophila Genetic Reference Panel (DGRP) [28, 34]. Additionally, the $MI_{24}$ is reduced by heterozygosity for a mutation of the NF-κB innate immune response transcription factor Relish [33]. Age and diet also play substantial roles in determining outcomes of TBI in flies. The $MI_{24}$ of flies injured at a younger age is lower than at an older age, and the $MI_{24}$ is lower for flies fed water versus high-carbohydrate diets during the 24 h following primary injuries [28, 29]. Furthermore, using the HIT device, Lee et al. (2019) demonstrated that β-hydroxybutyrate, a metabolite of KD, reduces TBI-induced aggression in flies [35]. Thus, to further explore the utility of the fly TBI model, we examined the effect of KD on the $MI_{24}$ and lifespan following TBI. We found that, relative to high-carbohydrate diets, high-fat KD reduced the $MI_{24}$ and increased lifespan following TBI and that

Eip75B, an ortholog of PPARγ, was necessary to mediate the beneficial effect of KD on the $MI_{24}$.

## Materials and methods

### Fly lines and culturing

Flies were maintained in humidified incubators at 25˚C on solid CMYD. DGRP lines and *Eip75B* mutant fly lines were obtained from the Bloomington Stock Center (Indiana University).

### Diets

Solid CMYD contains 30 g Difco granulated agar (Becton-Dickinson, Sparks, MD), 44 g YSC-1 yeast (Sigma, St. Louis, MO), 328 g cornmeal (Lab Scientific, Highlands, NJ), 400 ml unsulfured Grandma's molasses (Lab Scientific), 3.6 L water, 40 ml propionic acid (Sigma), and tegosept (8 g Methyl 4-hydroxybenzoate in 75 ml of 95% ethanol) (Sigma). YD contains YSC-1 yeast (Sigma) in water. KD is commercial mouse Teklad ketogenic diet (TD.96355) (Envigo) that contains 173.3 g/Kg casein, 2.6 g/Kg DL-methionine, 586.4 g/Kg vegetable shortening (Crisco), 86.2 g/Kg corn oil, 88.0 g/Kg cellulose, 13.0 g/Kg vitamin mix (Teklad 40060), 2.5 g/Kg choline bitartrate, 0.1 g/Kg tertiary butylhydroquinone (TBHQ), 20.0 g/Kg mineral mix (calcium phosphate deficient), 19.3 g/Kg dibasic calcium phosphate, 8.2 g/Kg calcium carbonate, and 0.4 g/Kg magnesium oxide. KD at 0.3 cal/200 μl was prepared by adding 1.1 g of Teklad ketogenic diet to 5 ml of water and stirring the solution for 1 min at about 95˚C. Table 1 provides the caloric contribution of carbohydrate, protein, and fat for each diet as well as the amount of each diet used to make 0.3 cal/200 μl solutions. Flies were fed water and diluted diets by placing 200 μl on a filter paper disc at the bottom of a vial.

### $MI_{24}$ and lifespan assays

Flies were injured using a HIT device as described by Katzenberger et al. [26, 27]. Vials containing 60 flies at 0–7 days old were injured by 4 strikes at 5 min intervals with the spring deflected to 90˚. Vials with mixed sex flies had approximately 30 males and 30 females. The Mortality Index at 24 h ($MI_{24}$) was calculated by subtracting the percent of uninjured flies that died from the percent of injured flies that died during the 24 h following TBI. The lifespan of adult flies that survived 24 h following TBI flies was determined using vials with 20 flies each. The number of surviving flies was counted daily until all flies had died. Flies were transferred to new vials approximately every 3 days. Flies were considered dead if they did not show obvious locomotor activity. Statistical analysis of survival by the Kaplan-Meier Fisher's Exact Test was performed using OASIS 2 (Online Application for Survival Analysis 2) [36].

**Table 1. Caloric content of diets used in Figs 1–4.**

| Diet | Percent calories from: | | | cal/g | 0.3 cal/200 μl |
|---|---|---|---|---|---|
| | Protein | Carbohydrate | Fat | | g diet/ml water |
| KD | 9.2 | 0.3 | 90.5 | 6.70 | 0.22 |
| CMYD | 4.9 | 92.7 | 2.4 | 3.31 | 0.45 |
| YD | 41.0 | 42.0 | 17.0 | 3.25 | 0.46 |
| Glucose | 0 | 100.0 | 0 | 3.74 | 0.40 |
| Sucrose | 0 | 100.0 | 0 | 3.94 | 0.38 |

KD, ketogenic diet; CMYD, cornmeal-molasses-yeast diet; YD, yeast diet.

## Results

### Ketogenic diet following TBI reduces the incidence of early mortality

We previously found that the diet consumed directly after TBI in *Drosophila* substantially affects the incidence of early mortality [28, 29]. Flies fed cornmeal-molasses-yeast diet (a standard laboratory fly diet) or simple carbohydrates (i.e., sucrose, glucose, and fructose) during the 24 h following TBI have a significantly higher $MI_{24}$ than flies fed water. To further explore the effect of diet on the $MI_{24}$, we examined different concentrations of cornmeal-molasses-yeast diet (CMYD), yeast diet (YD, *S. cerevisiae*), and ketogenic diet (KD, a commercial mouse ketogenic diet). Based on caloric content, CMYD is high in carbohydrate and low in protein and fat, YD is high in carbohydrate and protein and low in fat, and KD is high in fat and low in carbohydrate and protein (Table 1). Diets were dissolved in water at 0.5 g/ml, serially diluted by 2-fold increments in water down to 0.0625 g/ml, and 200 μl was absorbed onto a filter paper disc that was placed at the bottom of a vial. 0–7 day old, mixed sex $w^{1118}$ flies cultured on solid CMYD (i.e., undiluted CMYD), were subjected to four strikes from the HIT device with 5 min between strikes and transferred to vials with diets at different concentrations.

We found that the $MI_{24}$ increased to a similar extent with increasing concentrations of CMYD and YD (Fig 1A). In contrast, the $MI_{24}$ was not affected by increasing concentrations of KD. We also examined the effect of CMYD, YD, and KD as well as glucose and sucrose at approximately the same caloric content (0.3 cal/200 μl) (Table 1). $MI_{24}$ values were similar for flies fed CMYD, YD, glucose, or sucrose and they were significantly higher than the $MI_{24}$ of flies fed water (Fig 1B). In contrast, the $MI_{24}$ of flies fed KD was the same as that of flies fed water. These data support our prior finding that ingestion of carbohydrate after TBI increases the $MI_{24}$ and demonstrate that ingestion of fat after TBI does not increase the $MI_{24}$ compared with water.

An alternative interpretation of the data in Fig 1A and 1B is that flies did not consume KD, suggesting that starvation and water have equivalent effects on the $MI_{24}$. To test this possibility, we determined the $MI_{24}$ of 0–7 day old, mixed sex $w^{1118}$ flies that were starved by placing them in vials with a dry filter paper disc following TBI. In contrast with flies fed KD, the $MI_{24}$ of starved flies was substantially higher than that of flies fed water (Fig 1C), demonstrating that consuming KD rather than starvation was beneficial. As an additional approach to test if flies consumed KD, we examined the lifespan of uninjured, mixed sex $w^{1118}$ flies cultured on solid CMYD to 0–7 days old and thereafter on water or 0.3 cal/200 μl CMYD or KD. The median and maximum lifespans of flies cultured on KD (14.7 ± 1.1 days and 37 days, respectively) were longer than those of flies cultured on water (10.6 ± 0.1 days and 13 days, respectively) and shorter than those of flies cultured on CMYD (23.4 ± 0.6 days and 74 days, respectively), indicating that flies examined in Fig 1A and 1B consumed KD (Fig 1D). Further support for this conclusion is provided in Fig 4B.

### Ketogenic diet is similarly beneficial following TBI in females versus males and in different genetic backgrounds

To investigate whether KD has sex-specific effects on TBI outcomes, we compared effects of KD, water, and solid CMYD on the $MI_{24}$ of 0–7 day old female, male, and mixed sex $w^{1118}$ flies. In each case, solid CMYD resulted in a significantly higher $MI_{24}$ than both water and KD, and water and KD had equivalent $MI_{24}$ values (Fig 2A). Moreover, comparisons of male, female, and mixed sex flies, revealed that KD as well as solid CMYD and water had similar effects on the $MI_{24}$. Therefore, sex does not alter the effects of KD, water, and solid CMYD on secondary injury mechanisms that cause early mortality following TBI.

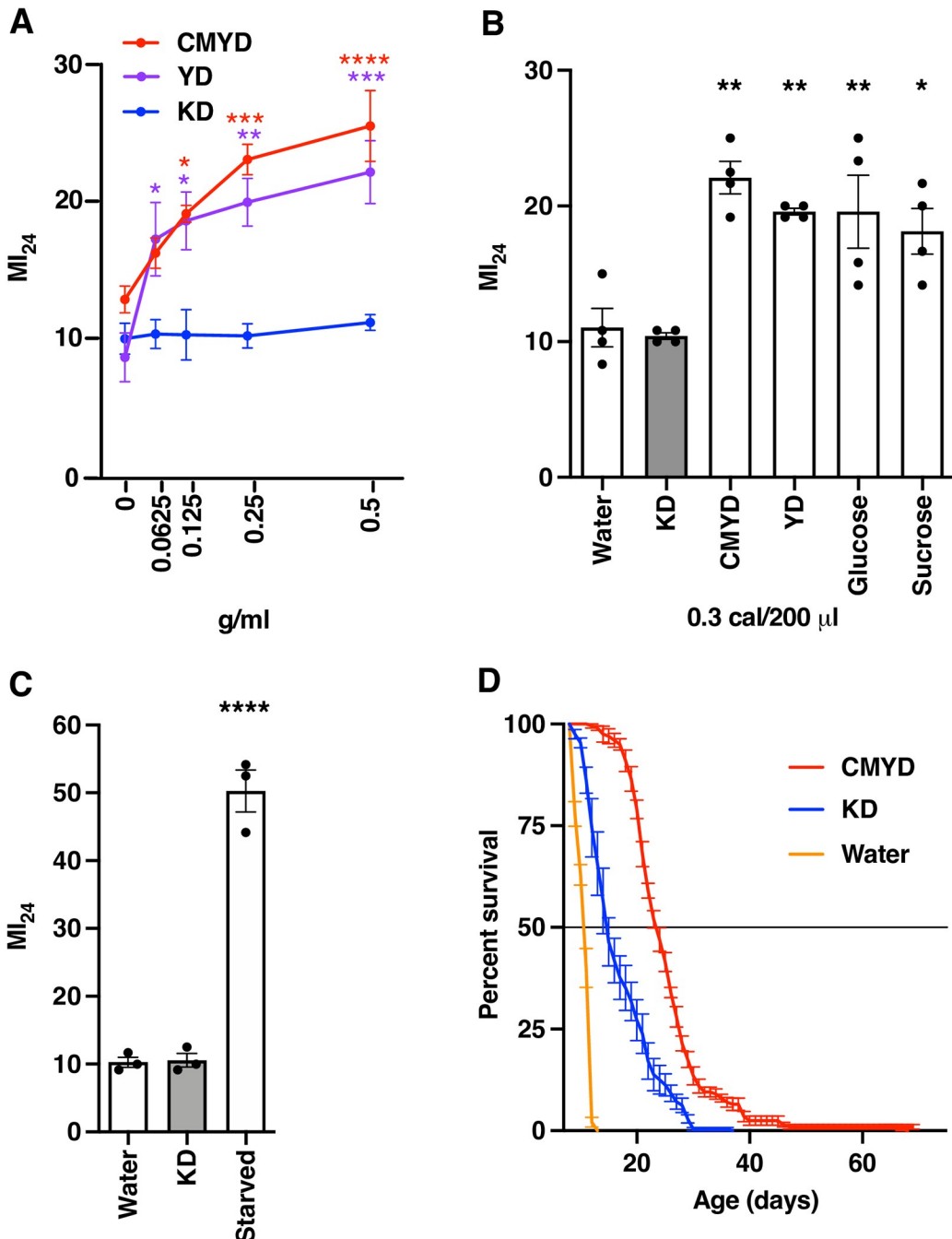

**Fig 1. Analysis of the effect of CMYD, YD, and KD on the MI$_{24}$.** (A) Dose-response analysis of the effect of CMYD, YD, and KD on the MI$_{24}$. The MI$_{24}$ represents the percent mortality of injured flies minus the percent mortality of uninjured flies 24 h following TBI. MI$_{24}$ values were determined for 0–7 day old, mixed sex $w^{1118}$ flies fed CMYD, YD, or KD at the indicated concentrations following TBI. At least three biological replicates of 60 flies were tested for each condition. Dots indicate the average MI$_{24}$, and error bars indicate the standard error of the mean (SEM). (B) MI$_{24}$ values were determined for 0–7 day old, mixed sex $w^{1118}$ flies fed water or CMYD, YD, KD, glucose, or sucrose at 0.3 cal/200 μl following TBI. Dots indicate biological replicates of 60 flies, bars indicate averages, and error bars indicate the SEM. (C) MI$_{24}$ values were determined for 0–7 day old, mixed sex $w^{1118}$ flies fed water, 0.3 cal/200 μl KD, or no food or water (starved) following TBI. Dots indicate biological replicates of 60 flies, bars indicate averages, and error bars indicate the SEM. (D) Percent survival was determined for uninjured 0–7 day old, mixed sex $w^{1118}$ flies fed water (n = 240) or 0.3 cal/200 μl CMYD (n = 200) or KD (n = 239) over the course of the experiment. Error bars indicate the SEM, and the horizontal line at 50% indicates the median lifespan. Significance for panels A, B, and C was determined using ordinary one-way ANOVA with Dunnett's Multiple Comparison test. $^{*}p<0.05$, $^{**}p<0.01$, $^{***}p<0.001$, and $^{****}p<0.0001$.

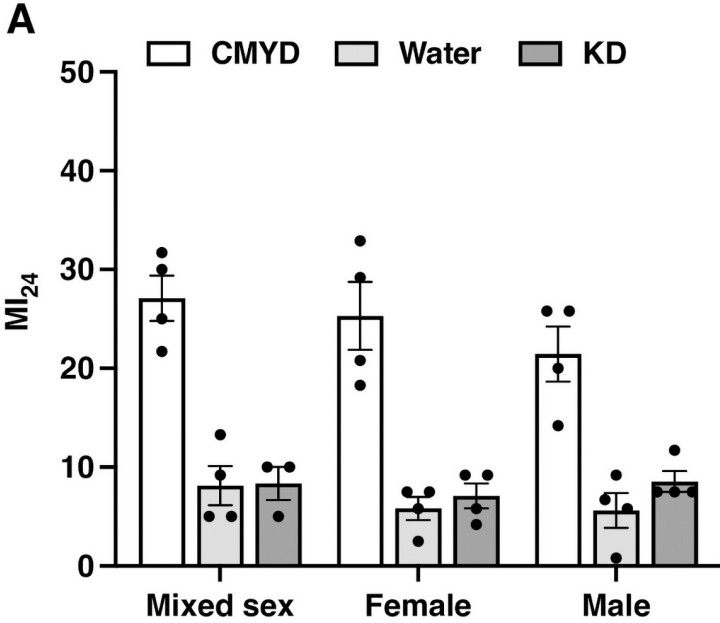

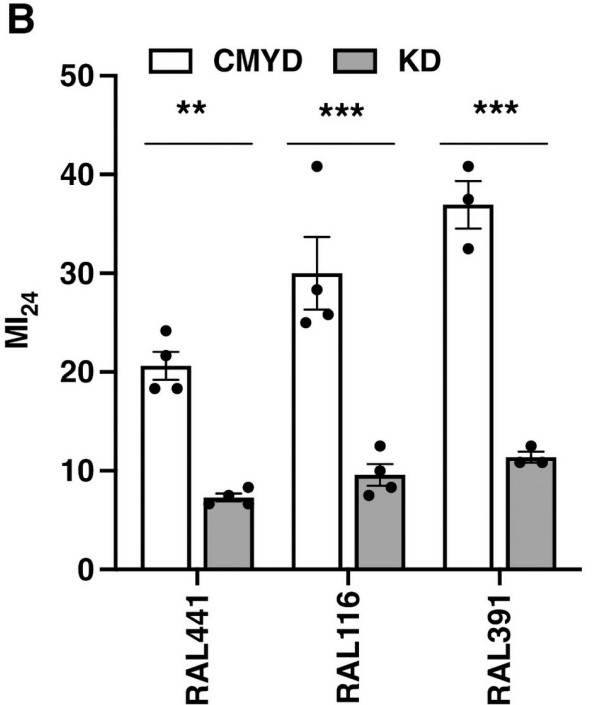

**Fig 2. The beneficial effect of KD on early mortality after TBI is equivalent in females and males and is conserved in different genetic backgrounds.** (A) $MI_{24}$ values were determined for 0–7 day old female, male, and mix sex $w^{1118}$ flies fed solid CMYD, water, or 0.3 cal/200 μl KD. The $MI_{24}$ represents the percent mortality of injured flies minus the percent mortality of uninjured flies 24 h following TBI. (B) $MI_{24}$ values were determined for 0–7 day old, mixed sex fly lines (RAL441, RAL116, and RAL391) from the DGRP fed 0.3 cal/200 μl CMYD or KD following TBI [34]. Dots indicate biological replicates of 60 flies, bars indicate averages, and error bars indicate the SEM. Significance was determined using ordinary one-way ANOVA with Dunnett's Multiple Comparison test. $^{**}p < 0.01$ and $^{***}p < 0.001$.

We previously found that the $MI_{24}$ of flies fed solid CMYD varied significantly among fly lines with different genetic backgrounds, including inbred fly lines from the Drosophila Genetic Reference Panel (DGRP) [26, 28, 29, 34]. To determine the extent to which genetic background affects the $MI_{24}$ of flies fed KD following TBI, we examined three lines from the DGRP that have different $MI_{24}$ values when fed solid CMYD following TBI [28, 29]. We fed 0–7 day old, mixed sex flies 0.3 cal/200 μl KD or CMYD for 24 h following TBI and determined the $MI_{24}$ of each line. For all three DGRP lines, the $MI_{24}$ of flies fed KD was lower than the $MI_{24}$ of flies fed CMYD (Fig 2B). Moreover, the $w^{1118}$ line and the DGRP lines fed KD had comparable $MI_{24}$ values, whereas these values varied among the same fly lines when fed CMYD (Figs 1B and 2B). These results indicate that the beneficial effect of KD on the $MI_{24}$ does not depend on the starting value of the $MI_{24}$ (on CMYD) in different fly lines, leading to uniformly low $MI_{24}$ values for flies fed KD. However, it does appear that the beneficial effect of KD has a limiting threshold beyond which it cannot act further, resulting in a proportionally greater rescuing effect for lines with higher $MI_{24}$ values on CMYD.

## Ketogenic diet following TBI has beneficial long-term effects on lifespan

For both humans and flies, individuals that survive TBI manifest a variety of long-term consequences, including reduced lifespan, as a result of secondary injuries triggered by primary injuries to the brain. The exact connection between primary injuries and secondary injuries is complex, and the details are still poorly understood. The fact that the $MI_{24}$ is reduced in flies fed KD immediately after TBI, raises the question of whether the beneficial effects of KD extend to longer-term pathological consequences of TBI. We examined this possibility using lifespan as a readout. Lifespan was determined for 0–7 day old, mixed sex $w^{1118}$ flies fed 0.3 cal/200 μl KD or CMYD for 24 h following TBI with surviving flies subsequently cultured on solid CMYD. As we reported previously, flies fed CMYD for 24 h after injury had a reduced lifespan relative to uninjured controls (Kaplan-Meier Fisher's Exact Test, $p = 4.1X10^{-9}$ at 50%) (Fig 3) [26, 29]. The same was true for flies fed KD (Kaplan-Meier Fisher's Exact Test, $p = 1.7X10^{-7}$ at 50%). However, notably, injured flies fed KD rather than CMYD for 24 h after injury had a significantly longer median lifespan (40.3 ± 0.2 days vs. 37.8 ± 0.28 days, Kaplan-Meier Fisher's Exact Test, $p = 1.0X10^{-4}$ at 50%). Moreover, the difference in median lifespan between injured flies and uninjured controls was much narrower for KD-fed than for CMYD-fed flies (40.3 ± 0.2 days vs. 44.2 ± 0.8 days for KD; 37.8 ± 0.28 days vs. 48.2 ± 1.0 days for CMYD). Thus, flies that avoid early mortality following TBI because of the beneficial effects of KD during a 24 h window after primary injuries continue to manifest long-term benefits of this diet weeks later.

## Beneficial effects of KD on early mortality are mediated by the PPARγ ortholog Eip75B

In mammals, the mechanism of action of KD is mediated by the transcription factor PPARγ, which has neuroprotective effects in a number of progressive neurological disorders, including TBI [19, 20]. Because KD exerts a protective effect following TBI in flies as well as mammals, we hypothesized that the underlying mechanism is conserved as well. If so, the protective effect in flies should depend on the transcription factor Eip75B (ecdysone-induced protein 75B), the *Drosophila* ortholog of PPARγ. The orthologous relationship is inferred both from amino acid sequence identity (i.e., Eip75B is the most significant match to human PPARγ in a BLAST search of the *Drosophila* proteome) and from common activation by the PPARγ agonist pioglitazone [37–39]. Under this hypothesis, mutational loss of *Eip75B* should result in loss of the beneficial effect of KD. Thus, we examined the effect of water and 0.3 cal/200 μl KD on the

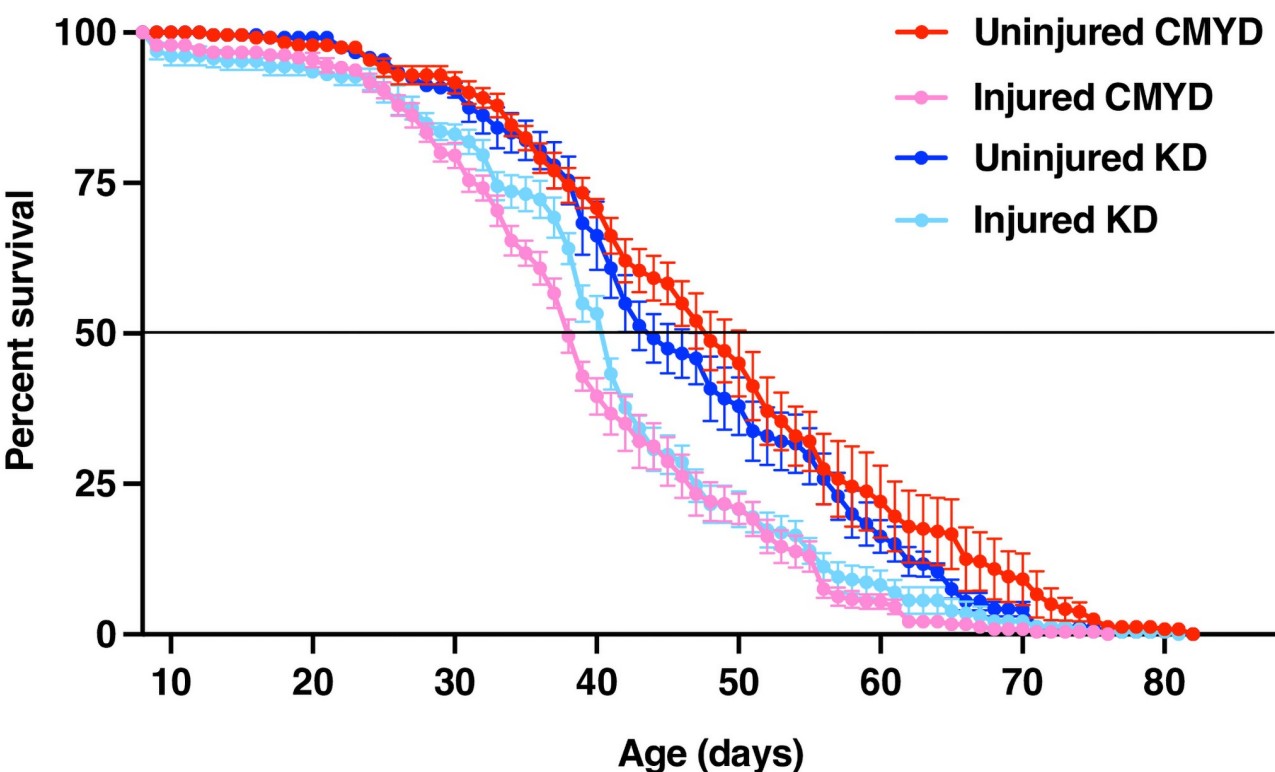

**Fig 3. KD has a long-term beneficial effect on lifespan following TBI.** Percent survival was determined for uninjured and injured 0–7 day old, mixed sex $w^{1118}$ flies fed 0.3 cal/200 µl CMYD or KD for 24 h following TBI and solid CMYD thereafter, that is, flies in the experiment that survived 24 h feeding on 0.3 cal/200 µl CMYD or KD were fed solid CMYD throughout the rest of their lifespan. At least 230 flies were examined for each condition. Error bars indicate the SEM, and the horizontal line at 50% indicates the median lifespan.

$MI_{24}$ of 0–7 day old, mixed sex *Eip75B* mutant flies. Three hypomorphic alleles of *Eip75B* (*Eip75B$^{MI04895}$*, *Eip75B$^{KG04491}$*, and *Eip75B$^{BG02576}$*) containing transposon insertions within the transcribed region were examined (Fig 4A). As in Fig 1, the $MI_{24}$ was comparably low in control $w^{1118}$ flies fed either water or KD (ordinary one-way ANOVA with Dunnett's Multiple Comparison test, $p = 0.835$) (Fig 4B). In contrast, although water-fed *Eip75B$^{KG04491}$*, *Eip75B$^{BG02576}$*, and *Eip75B$^{MI04895}$*/*Eip75B$^{BG02576}$* flies still had low $MI_{24}$ values comparable to that of water-fed $w^{1118}$ controls, $MI_{24}$ values were higher in KD-fed *Eip75B* mutants (ordinary one-way ANOVA with Dunnett's Multiple Comparison test, $p = 0.078$, $p = 0.033$, and $p = 0.006$, respectively), indicating that the beneficial effect of KD was impaired in these mutants. Furthermore, higher $MI_{24}$ values in KD-fed versus water-fed mutants provides further evidence that flies consumed KD. The beneficial effect of KD was, however, retained in *Eip75B$^{MI04895}$* homozygotes (ordinary one-way ANOVA with Dunnett's Multiple Comparison test, $p = 0.999$), which we attribute to a presumptive weaker loss of function of *Eip75B* caused by this mutation. *Eip75B$^{MI04895}$* only disrupts three of the seven *Eip75B* pre-mRNA isoforms, whereas *Eip75B$^{KG04491}$* and *Eip75B$^{BG02576}$* disrupt four and five isoforms, respectively (Fig 4A). Thus, while it remains possible that differences in genetic background underlie differences in $MI_{24}$ values for *Eip75B* mutant flies fed water versus KD, the data support the conclusion that activation of Eip75B/PPARγ by KD triggers mechanisms that reduce early mortality following TBI.

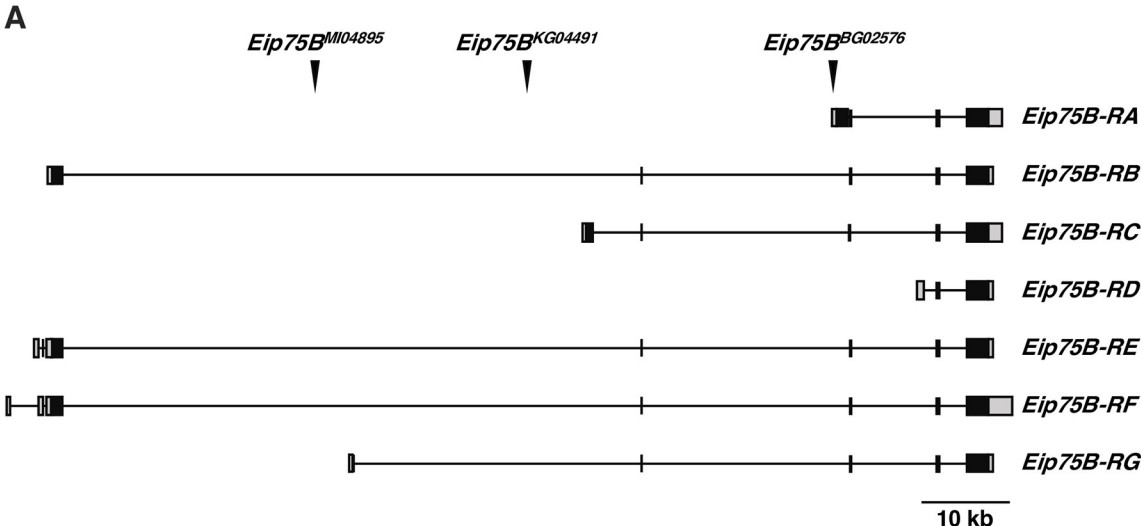

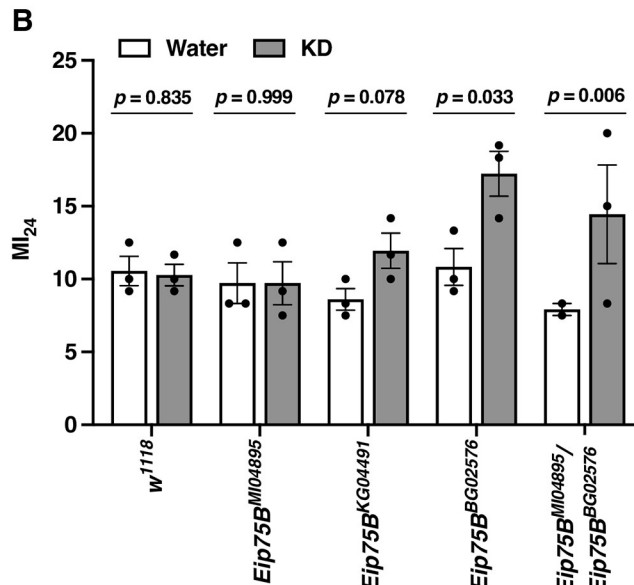

**Fig 4. The beneficial effect of KD on early mortality following TBI requires *Eip75B*.** (A) Transposon insertion locations (arrowheads) relative to the seven *Eip75B* transcripts drawn 5' to 3' (http://flybase.org). Gray boxes, black boxes, and lines indicate 5' and 3' untranslated regions, exons, and introns, respectively. (B) $MI_{24}$ values were determined for 0–7 day old, mixed sex $w^{1118}$ and *Eip75B* mutant flies fed water or 0.3 cal/200 μl KD following TBI. The $MI_{24}$ represents the percent mortality of injured flies minus the percent mortality of uninjured flies 24 h following TBI. Dots indicate biological replicates of 60 flies, bars indicate averages, and error bars indicate the SEM. Significance was determined using ordinary one-way ANOVA with Dunnett's Multiple Comparison test.

## Discussion

TBI patients face a spectrum of neurobehavioral sequelae initiated by primary injuries to the brain and mediated by the interplay of genetic and environmental factors that control patho-physiological cascades. Here, we discovered that an interaction between the genetic factor *Eip75B/PPARγ* and the environmental factor KD affects early mortality in a fly TBI model. In particular, whereas flies fed high-carbohydrate CMYD or YD exhibited a dose-dependent

increase in early mortality compared with flies fed water, flies fed high-fat KD showed no increase in early mortality (Fig 1A and 1B). The beneficial effect of KD on early mortality was equivalent in males and females, conserved in different genetic backgrounds, and had long-term beneficial effects on lifespan as well (Figs 2 and 3). However, the beneficial effect of KD on early mortality was diminished in flies mutant for Eip75B, a transcription factor ortholo-gous to mammalian PPARγ, suggesting that KD exerts its effect through Eip75B (Fig 4B). These data provide a mechanistic link between KD and PPARγ in modifying TBI outcomes and demonstrate the utility of the fly TBI model for dissecting interactions between genetic and environmental factors that affect TBI outcomes.

## The KD-Eip75B/PPARγ pathway may reduce early mortality following TBI by inhibiting Relish/NF-κB

Our data suggest that KD reduces early mortality following TBI by activating Eip75B/PPARγ. However, it remains to be determined what occurs downstream of Eip75B/PPARγ to exert this effect. One possibility is that Eip75B/PPARγ controls expression of genes involved in inflam-mation. In mammals, activation of PPARγ by dietary fatty acids mitigates neuroinflammation by inhibiting NF-κB, a transcriptional activator of cytokine, chemokine, and adhesion genes downstream of Toll-like receptor (TLR)/Interleukin-1 receptor (IL-1R) and Tumor necrosis factor-α receptor (TNFR) innate immune response signaling pathways [19]. PPARγ inhibits NF-κB by a variety of mechanisms, including ubiquitination and degradation, export from the nucleus, competition for cofactors, and steric inhibition of DNA binding [40]. In mammalian TBI models, reduced NF-κB activity resulting from treatment with the PPARγ agonist pioglita-zone or other pharmacological agents improves outcomes [22–25, 41–46]. Reducing NF-κB activity also improves TBI outcomes in flies [33]. Heterozygosity for a null mutation of *Relish*, one of three NF-κB genes in *Drosophila*, reduces early mortality and increases lifespan follow-ing TBI. Relish functions in the Immune-deficiency (Imd) pathway that is homologous to the TNFR pathway in mammals and controls transcription of numerous antimicrobial peptide genes (AMPs) that produce resistance to infection [47, 48]. A confirmed transcriptional target of Relish in TBI is the AMP gene *Metchnikowin* (*Mtk*), which when mutated reduces early mortality and increases lifespan following TBI [32]. Thus, KD-mediated activation of Eip75B/PPARγ may reduce early mortality following TBI by inhibiting Relish/NF-κB. This could be tested in the fly TBI model by examining effects of KD and pioglitazone on the $MI_{24}$, lifespan, and expression of AMP genes in wild type as well as *Relish* and *Mtk* mutant flies.

## KD and water appear to reduce early mortality following TBI by different mechanisms

Genetically diverse fly lines fed KD or water following TBI consistently had a lower incidence of early mortality relative to flies fed high-carbohydrate diets (Figs 1A and 1B and 2) [29]. These data suggest that KD and water might activate the same protective pathways. Water is a fasting condition where the amount of available carbohydrate is decreased, forcing a switch to the use of fatty acids as a nutrient supply through beta-oxidation and ketogenesis [49]. Keto-genesis converts acetyl-CoA to ketone bodies (e.g., β-hydroxybutyrate, acetone, and acetoace-tate) that are used by the brain and other tissues to produce energy. Flies with impaired mitochondrial ATP synthase activity produce elevated amounts of β-hydroxybutyrate, indicat-ing that ketogenesis operates in flies [50]. Additionally, aggressive behaviors and early mortal-ity induced by TBI in flies are reduced when flies are raised on high-carbohydrate diet supplemented with β-hydroxybutyrate relative to high-carbohydrate diet alone, indicating that β-hydroxybutyrate operates in the fly TBI model [35]. Nonetheless, several lines of evidence

argue that KD and water act by distinct mechanisms to reduce early mortality following TBI. First, the $MI_{24}$ of *Eip75B* mutants differed depending on whether they were fed KD or water, indicating that water acts independently of Eip75B/PPARγ (Fig 4B). Second, we previously found that flies fed water versus CMYD exhibited increased expression of AMP genes following TBI, suggesting that the beneficial effects of water are not mediated by inhibition of Relish, which activates the transcription of AMP genes [29]. Third, while heterozygosity for a mutation of *Relish* reduced the incidence of early mortality for flies fed CMYD following TBI, it did not affect the incidence of early mortality for flies fed water following TBI, suggesting that water functions either downstream or independently of Relish [33].

In conclusion, our observations indicate that KD signals through PPARγ to improve TBI outcomes in flies. Thus, the fly TBI model offers considerable potential for understanding the cellular and molecular mechanisms that underlie the beneficial effects of KD and may ultimately facilitate development of therapeutic intervention for TBI in humans.

## Acknowledgments

We thank members of the Boekhoff-Falk, Ganetzky, Perouansky, and Wassarman labs for helpful comments on this work.

## Author Contributions

**Conceptualization:** Joseph Blommer, Rebeccah J. Katzenberger, Barry Ganetzky, David A. Wassarman.

**Data curation:** Joseph Blommer, Megan C. Fischer, Rebeccah J. Katzenberger.

**Formal analysis:** Joseph Blommer, Megan C. Fischer, Athena R. Olszewski, Rebeccah J. Katzenberger, David A. Wassarman.

**Funding acquisition:** Barry Ganetzky, David A. Wassarman.

**Investigation:** Joseph Blommer, Megan C. Fischer, Athena R. Olszewski, Rebeccah J. Katzenberger.

**Methodology:** Joseph Blommer, Rebeccah J. Katzenberger.

**Project administration:** David A. Wassarman.

**Supervision:** Rebeccah J. Katzenberger, David A. Wassarman.

**Validation:** Joseph Blommer, Rebeccah J. Katzenberger.

**Visualization:** Joseph Blommer, Megan C. Fischer, Athena R. Olszewski, Rebeccah J. Katzenberger.

**Writing – original draft:** Joseph Blommer, Rebeccah J. Katzenberger, David A. Wassarman.

**Writing – review & editing:** Joseph Blommer, Megan C. Fischer, Athena R. Olszewski, Rebeccah J. Katzenberger, Barry Ganetzky, David A. Wassarman.

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
