## [Decision Letter · Decision Letter 0]

2 Aug 2021

PONE-D-21-21417

Ketogenic diet reduces early mortality following traumatic brain injury in Drosophila via the PPARg ortholog Eip75B

PLOS ONE

Dear Dr. Wassarman,

Thank you for submitting your manuscript to PLOS ONE. After careful consideration, we feel that it has merit but does not fully meet PLOS ONE’s publication criteria as it currently stands. Therefore, we invite you to submit a revised version of the manuscript that addresses the points raised during the review process.

Both reviewers and I concur that this is an interesting and timely manuscript. However there are a few questions by both reviewers that need to be addressed as outlined below and aim to clarify and improve the manuscript 

We look forward to receiving your revised manuscript.

Kind regards,

Efthimios M. C. Skoulakis, PhD

Academic Editor

PLOS ONE

Journal Requirements:

[This work was supported by NIH grant RF1 NS114359 to BG and DAW. JB was supported by a Sophomore Research Fellowship and a Hilldale Research Fellowship from UW-Madison. MCF was supported by a UW-Madison Genetics Department Summer Fellowship.]

 [The funders had no role in study design, data collection and analysis, decision to publish, or preparation of the manuscript.]

Reviewers' comments:

Reviewer's Responses to Questions

**Comments to the Author**

1. Is the manuscript technically sound, and do the data support the conclusions?

Reviewer #1: Partly

Reviewer #2: Yes

2. Has the statistical analysis been performed appropriately and rigorously? 

Reviewer #1: Yes

Reviewer #2: Yes

3. Have the authors made all data underlying the findings in their manuscript fully available?

Reviewer #1: Yes

Reviewer #2: Yes

4. Is the manuscript presented in an intelligible fashion and written in standard English?

Reviewer #1: Yes

Reviewer #2: Yes

5. Review Comments to the Author

Reviewer #1: In this manuscript the authors test the impact of feeding flies a ketogenic (KD) vs high carbohydrate diet (CMYD) in the 24 hrs after TBI. The system they use is the so-called HIT device to administer TBI. They find that animals on a the CMYD diet or a yeast diet have standard high mortality after 24 hrs, whereas animals on a KD have low mortality similar to feeding animals water. They provide evidence that the animals are eating because starved animals have a high mortality. They show that the protective effect of a KD is conferred upon a range of genetic backgrounds, and then provide molecular insight by showing that the effect is dependent on Eip75B gene function, which is the homolog of PPAR-gamma. These findings suggest that KD, by activating Eip75B activity, mitigates the deleterious outcomes of TBI. These data are consistent with some mammalian findings.

Overall this is an interesting and appropriate manuscript. There are just a few points that need to be addressed..

In Figure 2, are the lifespans of the different strains on KD significantly different? The RAL441 and RAL391 look like they might be. Thus, whereas on a CMYD diet, the mortality is marked different, on a KD diet it may still be different but overall dramatically better.

In figure 3, at first it is rather confusing why these lifespans are so dramatically different from those in figure 1. They could clarify this by making the point more clearly that the flies are cultured differently. That is…

”Lifespan was determined…for 24 h following TBI but then surviving flies were cultured on solid CMYD.”

Also, some of their arguments about lifespan (median lifespan between injured and uninjured animals; Fig 3) seem sketchy because the uninjured CMYD animals have a better median lifespan than KD animals. So the animals survive better after TBI if on a KD for 24 hrs, but after that, the KD lifespan is compromised. So it seems like there are pros and cons. How do they explain or interpret this.

Since only 1 of two Eip75B alleles showed loss of the KD benefit, the significance of Eip75B function seems questionable also. Maybe those results are due to genetic background. Can they show somehow that the alleles are of a strength consistent with their interpretation that the MI04895 allele is less severe? This needs another allele, and/or controls for background, or some other way of validating the findings to be properly interpreted. Perhaps they could measure activation of PPAR-gamma to at least show correlation. In any case, this point needs to be strengthened.

Reviewer #2: This is an interesting paper by Blommer et al that provides novel experimental data indicating that a ketogenic diet can mediate beneficial effects in a fly model of traumatic brain injury. My comments are as followed:

1. Please make sure that gene names are consistently italicized.

2. According to the Alliance of Genome Resources, Eip75B is an ortholog of the following human genes: NR1D2, PPARA, PPARG, NR1D1, and PPARD. It may be worthwhile to mention that PPARG is not the sole ortholog of this fly gene.

3. In the Introduction, I would mention that traumatic brain injury is a known risk factor dementia. This will help highlight the broader importance of your research. There are multiple papers one could cite for this, such as this recent systematic review and meta-analysis (PMID: 33044182).

4. In the Introduction, it may be useful to mention that there is a growing interest in the interplay between lipid metabolism and aging. Dr. Anne Brunet has published excellent reviews on this topic.

5. It would be helpful if the authors can explicitly list the increase in median and mean lifespan in response to a ketogenic diet. For those looking to do systematic reviews and/or meta-analyses in the future, this information could be useful. It may be simplest to do this in the form of a table that summarizes all of the lifespan results (i.e., p-value, median increase, mean increase).

6. While there is evidence that a ketogenic diet may be beneficial in specific circumstances, it may not be as impactful as other dietary interventions (e.g., caloric restriction, intermittent fasting, Mediterranean diet, plant-based diet). In the Discussion, can the authors comment on where a ketogenic diet will be specifically beneficial vs. other dietary alterations and why they think this is the case?

7. Major sections vs. sub-sections need to be demarcated more clearly. I recommend making major sections entirely capitalized (e.g., DISCUSSION) while keeping sub-sections as sentence case (e.g., KD and water appear to reduce early mortality following TBI by different mechanisms).

8. Unless the journal handles this separately, please add a Conflict of Interest section to your manuscript.

6. PLOS authors have the option to publish the peer review history of their article (what does this mean?). If published, this will include your full peer review and any attached files.

Reviewer #1: No

Reviewer #2: No

---

## [Author Response · Author response to Decision Letter 0]

9 Sep 2021

Responses to reviewer’s criticisms (in bold font)

Reviewer #1: In this manuscript the authors test the impact of feeding flies a ketogenic (KD) vs high carbohydrate diet (CMYD) in the 24 hrs after TBI. The system they use is the so-called HIT device to administer TBI. They find that animals on a the CMYD diet or a yeast diet have standard high mortality after 24 hrs, whereas animals on a KD have low mortality similar to feeding animals water. They provide evidence that the animals are eating because starved animals have a high mortality. They show that the protective effect of a KD is conferred upon a range of genetic backgrounds, and then provide molecular insight by showing that the effect is dependent on Eip75B gene function, which is the homolog of PPAR-gamma. These findings suggest that KD, by activating Eip75B activity, mitigates the deleterious outcomes of TBI. These data are consistent with some mammalian findings.

Overall this is an interesting and appropriate manuscript. There are just a few points that need to be addressed.

In Figure 2, are the lifespans of the different strains on KD significantly different? The RAL441 and RAL391 look like they might be. Thus, whereas on a CMYD diet, the mortality is marked different, on a KD diet it may still be different but overall dramatically better.

Previously, we found that fly lifespan affects the MI24, that is, at a given age, flies with a short lifespan have a higher MI24 than flies with a long lifespan (Katzenberger et al. (2013) PNAS). Thus, it is likely that RAL441 flies have a longer lifespan than RAL391 flies when fed solid CMYD. We agree that it would be interesting to determine if the lifespan of these lines differs when fed KD, but we believe that this is beyond the scope of the manuscript. 

In figure 3, at first it is rather confusing why these lifespans are so dramatically different from those in figure 1. They could clarify this by making the point more clearly that the flies are cultured differently. That is…

”Lifespan was determined…for 24 h following TBI but then surviving flies were cultured on solid CMYD.”

We clarified this point in several ways. In the legend for Figure 1, we added the phrase “over the course of the experiment” to the sentence “Percent survival was determined for uninjured 0-7 day old, mixed sex w1118 flies fed water (n=240) or 0.3 cal/200 �l CMYD (n=200) or KD (n=239) over the course of the experiment.” Also, in the legend for Figure 3, we added the phrase “that is, flies in the experiment that survived 24 h feeding on 0.3 cal/200 �l CMYD or KD were fed solid CMYD throughout the rest of their lifespan” to the sentence “Percent survival was determined for uninjured and injured 0-7 day old, mixed sex w1118 flies fed 0.3 cal/200 �l CMYD or KD for 24 h following TBI and solid CMYD thereafter, that is, flies in the experiment that survived 24 h feeding on 0.3 cal/200 �l CMYD or KD were fed solid CMYD throughout the rest of their lifespan.”

Also, some of their arguments about lifespan (median lifespan between injured and uninjured animals; Fig 3) seem sketchy because the uninjured CMYD animals have a better median lifespan than KD animals. So the animals survive better after TBI if on a KD for 24 hrs, but after that, the KD lifespan is compromised. So it seems like there are pros and cons. How do they explain or interpret this.

We see the reviewer’s point that the data are not black and white. But we think that our main conclusion is supported by the data. Figure 3 shows that the survival curve of injured KD flies more nearly approximates that of uninjured KD flies than does that of injured CYMD flies to their control. Thus, relative to the appropriate control, the ketogenic diet is beneficial both at 24 hours and throughout the course of the lifespan for flies that survived mortality at 24 hours. Accordingly, in the Abstract we conclude “flies protected from early mortality by KD continued to show survival benefits weeks later” and in the section about Figure 3, we similarly conclude “Thus, flies that avoid mortality following TBI because of the beneficial effects of KD during a 24 h window after primary injuries continue to manifest long-term benefits of this diet weeks later.”

Since only 1 of two Eip75B alleles showed loss of the KD benefit, the significance of Eip75B function seems questionable also. Maybe those results are due to genetic background. Can they show somehow that the alleles are of a strength consistent with their interpretation that the MI04895 allele is less severe? This needs another allele, and/or controls for background, or some other way of validating the findings to be properly interpreted. Perhaps they could measure activation of PPAR-gamma to at least show correlation. In any case, this point needs to be strengthened.

To address this point, we examined another Eip75B allele (Eip75BKG04491) and included these data in Figure 4. Thus, there are now three Eip75B alleles or allelic combinations in which the MI24 is higher for KD than water. We also added a figure (Figure 4A) that supports the conclusion that Eip75BMI04895 is a weaker allele than Eip75BKG04491 and Eip75BBG02576. Lastly, we added analyses of male and female flies to Figure 2 (new panel 2A) that reinforce the finding that water and KD have similar effects on the MI24 of flies that are wild type for Eip75B. Based on these data, we have concluded that “The beneficial effect of KD was, however, retained in Eip75BMI04895 homozygotes (ordinary one-way ANOVA with Dunnett’s Multiple Comparison test, p=0.999), which we attribute to a presumptive weaker loss of function of Eip75B caused by this mutation. Eip75BMI04895 only disrupts three of the seven Eip75B pre-mRNA isoforms, whereas Eip75BKG04491 and Eip75BBG02576 disrupt four and five isoforms, respectively (Fig. 4A). Thus, while it remains possible that differences in genetic background underlie differences in MI24 values for Eip75B mutant flies fed water versus KD, the data support the conclusion that activation of Eip75B/PPAR� by KD triggers mechanisms that reduce early mortality following TBI.”

Reviewer #2: This is an interesting paper by Blommer et al that provides novel experimental data indicating that a ketogenic diet can mediate beneficial effects in a fly model of traumatic brain injury. My comments are as followed:

1. Please make sure that gene names are consistently italicized.

We believe that all of the gene names are italicized. In accord with Drosophila convention, genes and RNAs are italicized, and proteins are not italicized.

2. According to the Alliance of Genome Resources, Eip75B is an ortholog of the following human genes: NR1D2, PPARA, PPARG, NR1D1, and PPARD. It may be worthwhile to mention that PPARG is not the sole ortholog of this fly gene.

Our assignment of Eip75B as the ortholog of PPAR� is based on two lines of evidence. First, BLAST search analysis of the Drosophila genome with the human PPAR��protein identified Eip75B as the most significant match. Second, data in references 38 and 39 demonstrate that Eip75B functions similarly to PPAR Thus, we have added the sentence “The orthologous relationship is inferred both from amino acid sequence identity (i.e., Eip75B is the most significant match to human PPAR� in a BLAST search of the Drosophila proteome).” This information is provided under the section heading “Beneficial effects of KD on early mortality are mediated by the PPAR� ortholog Eip75B.” PPARA, PPARD, NR1D1, and NR1D2 may be listed as orthologs simply because they are similar in sequence to Eip75B. Often single genes in flies are represented by expanded gene families in mammals.

3. In the Introduction, I would mention that traumatic brain injury is a known risk factor dementia. This will help highlight the broader importance of your research. There are multiple papers one could cite for this, such as this recent systematic review and meta-analysis (PMID: 33044182).

We appreciate the suggestions to include dementia and the interplay between lipid metabolism and aging in the Introduction. Our rationale for not including these topics as well as many others related to TBI is to focus the Introduction on topics that are critical for the reader to understand and to appreciate the significance and implications of the data that are presented. 

4. In the Introduction, it may be useful to mention that there is a growing interest in the interplay between lipid metabolism and aging. Dr. Anne Brunet has published excellent reviews on this topic.

Please see the response to point 3.

5. It would be helpful if the authors can explicitly list the increase in median and mean lifespan in response to a ketogenic diet. For those looking to do systematic reviews and/or meta-analyses in the future, this information could be useful. It may be simplest to do this in the form of a table that summarizes all of the lifespan results (i.e., p-value, median increase, mean increase).

For readability, we have provided median lifespan numbers and p-values in the text as they were discussed. 

6. While there is evidence that a ketogenic diet may be beneficial in specific circumstances, it may not be as impactful as other dietary interventions (e.g., caloric restriction, intermittent fasting, Mediterranean diet, plant-based diet). In the Discussion, can the authors comment on where a ketogenic diet will be specifically beneficial vs. other dietary alterations and why they think this is the case?

Unfortunately, there are very little data in the mammalian literature and no data in the Drosophila literature on the effect of other dietary interventions on TBI. Thus, we do not feel able to add any meaningful comparisons among the various diets at this time.

7. Major sections vs. sub-sections need to be demarcated more clearly. I recommend making major sections entirely capitalized (e.g., DISCUSSION) while keeping sub-sections as sentence case (e.g., KD and water appear to reduce early mortality following TBI by different mechanisms).

The guidelines for style are set by PLoS One, which we are required to follow. 

8. Unless the journal handles this separately, please add a Conflict of Interest section to your manuscript.

Please see the response to point 7.

---

## [Decision Letter · Decision Letter 1]

20 Sep 2021

PONE-D-21-21417R1Ketogenic diet reduces early mortality following traumatic brain injury in Drosophila via the PPARg ortholog Eip75BPLOS ONE

Dear Dr. Wassarman,

Thank you for submitting your manuscript to PLOS ONE. After careful consideration, we feel that it has merit but does not fully meet PLOS ONE’s publication criteria as it currently stands. Therefore, we invite you to submit a revised version of the manuscript that addresses the points raised during the review process.

the reviewers feel and I concur that the manuscript is ready for publication save for the clarifications presented below. It is essential in my opinion to clarify the MI24 measurements are indeed %.Please make these minor changes and submit ASAP.

We look forward to receiving your revised manuscript.

Kind regards,

Efthimios M. C. Skoulakis, PhD

Academic Editor

PLOS ONE

Journal Requirements:

Reviewers' comments:

Reviewer's Responses to Questions

**Comments to the Author**

1. If the authors have adequately addressed your comments raised in a previous round of review and you feel that this manuscript is now acceptable for publication, you may indicate that here to bypass the “Comments to the Author” section, enter your conflict of interest statement in the “Confidential to Editor” section, and submit your "Accept" recommendation.

Reviewer #1: All comments have been addressed

Reviewer #2: (No Response)

2. Is the manuscript technically sound, and do the data support the conclusions?

Reviewer #1: Yes

Reviewer #2: Yes

3. Has the statistical analysis been performed appropriately and rigorously? 

Reviewer #1: Yes

Reviewer #2: Yes

4. Have the authors made all data underlying the findings in their manuscript fully available?

Reviewer #1: Yes

Reviewer #2: Yes

5. Is the manuscript presented in an intelligible fashion and written in standard English?

Reviewer #1: Yes

Reviewer #2: Yes

6. Review Comments to the Author

Reviewer #1: This manuscript has been improved by the revisions suggested by the Reviewers.

One minor point is to make clear along the y-axes that MI24 is a percentage. It is unlabelled and not clarified in the legends.

Also in the text on page on page 4, "varies from 7 to 58" should be "7% to 58% among 179...."

Reviewer #2: It is surprising that the authors were unwilling to accommodate minor changes to the text that would have helped emphasize the importance of this work to a broader audience. While my recommendation is to accept the paper based on its technical soundness and its improvement from the review process, I would encourage the authors to be more flexible in the future.

7. PLOS authors have the option to publish the peer review history of their article (what does this mean?). If published, this will include your full peer review and any attached files.

Reviewer #1: No

Reviewer #2: No

---

## [Author Response · Author response to Decision Letter 1]

21 Sep 2021

Responses to reviewer’s criticisms 

Reviewer #1: This manuscript has been improved by the revisions suggested by the Reviewers.

One minor point is to make clear along the y-axes that MI24 is a percentage. It is unlabelled and not clarified in the legends.

Also in the text on page on page 4, "varies from 7 to 58" should be "7% to 58% among 179...."

We appreciate the reviewer’s suggestion. To increase clarity, we added the following sentence to the legends for Figures 1, 2, and 4. “The MI24 represents the percent mortality of injured flies minus the percent mortality of uninjured flies 24 h following TBI.”

Adding percent or % to the y-axis label is problematic because the MI24 is defined as a percent. So, MI24 percent and MI24% are redundant and percent mortality and % mortality are incorrect. Furthermore, I would like the graphs to be consistent with our prior publications. We have published graphs in nine papers with y-axes labeled MI24, and other labs have also used this nomenclature. 

Reviewer #2: It is surprising that the authors were unwilling to accommodate minor changes to the text that would have helped emphasize the importance of this work to a broader audience. While my recommendation is to accept the paper based on its technical soundness and its improvement from the review process, I would encourage the authors to be more flexible in the future.

We thank the reviewer for the suggestion.

---

## [Editor Report · Decision Letter 2]

7 Oct 2021

Ketogenic diet reduces early mortality following traumatic brain injury in Drosophila via the PPARg ortholog Eip75B

PONE-D-21-21417R2

Dear Dr. Wassarman,

We’re pleased to inform you that your manuscript has been judged scientifically suitable for publication and will be formally accepted for publication once it meets all outstanding technical requirements.

Kind regards,

Efthimios M. C. Skoulakis, PhD

Academic Editor

PLOS ONE
---

## [Editor Report · Acceptance letter]

15 Oct 2021

PONE-D-21-21417R2 

Ketogenic diet reduces early mortality following traumatic brain injury in * Drosophila * via the PPARg ortholog Eip75B 

Dear Dr. Wassarman:

I'm pleased to inform you that your manuscript has been deemed suitable for publication in PLOS ONE. Congratulations! Your manuscript is now with our production department. 

Kind regards, 

on behalf of

Dr. Efthimios M. C. Skoulakis 

Academic Editor

PLOS ONE